# 𝒫𝒯 Symmetry, Non-Gaussian Path Integrals, and the Quantum Black–Scholes Equation

**DOI:** 10.3390/e21020105

**Published:** 2019-01-23

**Authors:** Will Hicks

**Affiliations:** Investec Bank PLC, 30 Gresham Street, London EC2V 7QP, UK; whicks7940@googlemail.com

**Keywords:** quantum Black–Scholes, non-Gaussian kernels, quantum stochastic calculus, 𝒫𝒯 symmetric quantum mechanics

## Abstract

The Accardi–Boukas quantum Black–Scholes framework, provides a means by which one can apply the Hudson–Parthasarathy quantum stochastic calculus to problems in finance. Solutions to these equations can be modelled using nonlocal diffusion processes, via a Kramers–Moyal expansion, and this provides useful tools to understand their behaviour. In this paper we develop further links between quantum stochastic processes, and nonlocal diffusions, by inverting the question, and showing how certain nonlocal diffusions can be written as quantum stochastic processes. We then go on to show how one can use path integral formalism, and PT symmetric quantum mechanics, to build a non-Gaussian kernel function for the Accardi–Boukas quantum Black–Scholes. Behaviours observed in the real market are a natural model output, rather than something that must be deliberately included.

## 1. Introduction

There are several different approaches to modelling the random nature of the financial markets. The traditional method, most used by practitioners in the finance industry, involves the application of Brownian motion and Ito calculus (for example see [1]). This leads to parabolic partial differential equations, such as the Black–Scholes equation, that can be solved to derive the price for derivative contracts.

The application of quantum formalism to Mathematical Finance has been investigated by several sources. For example in [2,3], Haven discusses the implications of modelling the price of a derivative security as a state function in a Schrödinger equation.

Furthermore, in [4] Baaquie, and in [5] Linetsky, they apply the path integral formulation to financial modelling, and show how the standard Black–Scholes equation can be derived from a simple Lagrangian. In [6,7,8], Baaquie goes on to discuss the implication of different potential terms, as well as the implications of including acceleration terms in the relevant Hamiltonian.

In [9], Segal and Segal discuss the dual approach of viewing the price of a derivative security as an observable rather than a state. In [10], Accardi and Boukas use the Hudson–Parthasarathy calculus described in [11], to develop this idea, and derive the relevant Black–Scholes equations.

Despite drawing from quantum formalism, the Accardi–Boukas method is, in some respects, philosophically close to the classical approach to finance. Both the classical approach, and the Accardi–Boukas approach, use the no-arbitrage and self-financing assumptions to derive the relevant partial differential equation (PDE). In the classical case, the randomness is introduced via an Ito stochastic process, while in the Accardi–Boukas case a Hudson–Parthasarathy quantum stochastic process is used. In fact, as is discussed in Section 2, the Accardi–Boukas framework incorporates aspects of both classical and quantum approaches. The intrinsic uncertainty of the current market can be incorporated into an initial market quantum state, while the dynamics going forward are incorporated into the quantum stochastic process for the observable (for example tradeable Financial Times Stock Exchange, or FTSE, price).

In [12], it is shown that the Quantum Black–Scholes equation of Accardi–Boukas can be modelled as a nonlocal diffusion process, and that solutions can be found using McKean–Vlasov stochastic differential equations ([13]). The particle method (for example see [14]) is then applied to the simulation of solutions.

In this paper, we start in Section 2 with an introduction to the Accardi–Boukas quantum Black–Scholes equation. This is intended to introduce the techniques of quantum stochastic calculus, and the focus is on key principals and applications. We refer to [10,11], for technical details. Readers already familiar can skip to Section 3.

Given the results outlined in [12], it is natural to ask whether there is a deeper link between nonlocal diffusions, and quantum stochastic processes. In Section 3 we aim to take the first step in answering this question by inverting the problem considered. Rather than starting from a specific quantum stochastic process and writing the solution as a nonlocal diffusion, we show that given a nonlocal diffusion, in certain cases it is possible to associate an equivalent quantum stochastic process. In [15], Henry-Labordère shows that the local volatility pricing model, originally developed by Dupire in [16], can be modelled using a diffusion on a Riemannian manifold. We solve the problem in hand by extending this idea to a *nonlocal* diffusion on a Riemannian manifold.

In Section 4, we seek to develop the path integral approach associated with the Accardi–Boukas quantum Black–Scholes equations. Using this method one can derive the partial differential equations of finance using least action principals, in a way that is consistent with the basic principals of quantum physics. In addition to providing an alternative theoretical viewpoint from which to understand a system, this also yields numerical techniques that can be applied.

Attempting to write down the quantum Kolmogorov backward equation as a Schrödinger equation leads to a non-Hermitian Hamiltonian function, that is an infinite power series in the momentum variable. In Section 4.2, we show how the PT symmetry condition, discussed by Bender in [17], applies in this case. The analysis presented in [17] shows PT symmetry is sufficient to ensure that this will still lead to a valid quantum theory. We then show how extending the usual Wick rotation to both the time and space variables enables us to write the quantum Kolmogorov backward equation as a Schrödinger equation as required.

In Section 4.4, we follow standard steps in deriving the relevant non-Gaussian kernel using the Fourier transform (see for example Chapter 20 of [18]). In Section 4.5, we use the alternative Legendre transform method again following standard steps that have previously been used to derive the Wiener measure, and Gaussian kernels (for example see [19]). In addition to giving more physical insight, this method leads to an expression for an integral kernel, that can be used in Monte-Carlo simulations. Common market behaviours such as volatility skew, and fat tails, occur completely naturally as a result of the Hamiltonian function, in a way that is not observed when carrying out the equivalent derivation for the Wiener measure more commonly used for financial modelling. In the classical case, such behaviours require additional model features to be added, such as stochastic volatility.

## 2. Quantum Stochastic Processes and the Accardi–Boukas Quantum Black–Scholes

A classical *k*-dimensional random variable can be defined as mapping from a probability space, Ω to a real vector in $R^{k}$. The domain for the mapping is controlled by a sigma algebra F.

A classical stochastic process is a random variable, indexed by an interval of the real line (time). $X:\Omega \times T\to R^{k}$. Now the domain is controlled by a filtration: $F_{t}$, where: $F_{s}\subset F_{t}$ if s<t. See for example, [1].

In fact, the full technical detail of probability theory (while important) can be ignored for most practical purposes in finance. The key building blocks are the Wiener process: W(t) (which can be thought of as a real valued function on the real line), the Ito integral (the existence of which can be taken as given), and Ito’s lemma.

We take the same approach in this section: the full mathematical machinery behind quantum field theory, and the construction of path integrals, is unnecessary for practical purposes in finance. Readers familiar with the Hudson–Parthasarathy quantum stochastic calculus ([11]), and the Accardi–Boukas Quantum Black–Scholes ([10]), can skip to Section 3.

### 2.1. Market State Space

The market that we are trying to model can be described by a state function sitting in the tensor product of the initial space H and the Boson Fock space: $\Gamma (L^{2}\left(R^{+},H\right))$. This is described below.

#### 2.1.1. Initial Space

The initial space: H, is a Hilbert space that carries the price information from the current market. If we want to know the current price of the FTSE index, then this is represented by the operator *X*, where *X* acts on the state function ϕ(x) by pointwise multiplication: Xϕ=xϕ(x). To get the expected price one can trade the FTSE index at right now, we carry out the following calculation:$$E\left[X\right]=\langle \phi ,X\phi \rangle =\int_{R}x{\left|\phi \left(x\right)\right|}^{2}dx,\mathrm{for}\;\phi \left(x\right)\in H.$$

If I know with certainty, that the FTSE is at 7000, then the initial state function would be a Dirac state: ${\left|\phi \left(x\right)\right|}^{2}=\delta \left(7000-x\right)$.

#### 2.1.2. Boson Fock Space

To define a quantum stochastic process we require a mechanism to incrementally amend the initial quantum state as time progresses, essentially by adding the drift and the random diffusion. This is achieved using the Boson Fock space.

We start with functions from the time axis, with values in the Hilbert space that carries the pricing information (H). This space is written: $K=L^{2}\left(R^{+},H\right)$.

Next we take the exponential vectors, ψ(f). For f∈K we have: $\psi \left(f\right)=\left(1,f,\frac{f⊗f}{\sqrt{2}},\ldots ,\frac{f^{⊗n}}{n^{1/2}},\ldots \right)$, so that: $\langle \psi \left(f\right),\psi \left(g\right)\rangle =e^{\langle f,g\rangle }$, for f,g∈K. The Boson Fock space is defined as the Hilbert space completion of these exponential vectors, which now provide the mechanism we require.

Our market state space is the tensor product space: H⊗Γ(K). Initially at t=0, the Boson Fock space can be thought of as being empty (although this turns out to be unimportant). The operator *X* that returns the expected FTSE price becomes X⊗I, where I represents the identity operator on the empty Boson Fock space, and the calculation of the FTSE index price right now, is unchanged.

#### 2.1.3. Quantum Drift

A particle, with initial wave function $\phi \left(x\right)=\int_{R}\tilde{\phi }\left(p\right)e^{ipx}dp$, in a system controlled by the Hamiltonian function H(x,p), where *p* represents the momentum, has a unitary time development operator: $U_{t}=e^{iHt}$. Thus if the operator $X_{0}$ returned the position at time 0, then we have at time *t* (we assume Planck’s constant ℏ=1):(1)$$X_{t}=j_{t}\left(X_{0}\right)=U_{t}^{*}X_{0}U_{t}$$

The Hamiltonian function H(x,p) is the infinitesimal generator for the time development operator, and we can write the following quantum stochastic differential equation (SDE):(2)$$dU_{t}=\left(-iHdt\right)U_{t}$$

The situation for modelling the FTSE is exactly the same. To define a quantum SDE with drift, we require a self-adjoint operator *H*, which controls the drift through Equations (Equation 1) and (Equation 2).

For a classical particle with drift, the position is a deterministic function of time. Now the position of the particle is no longer deterministic. It is the wave function that evolves in a deterministic fashion.

#### 2.1.4. Quantum Diffusion

We now add operators that allow the market state function to evolve stochastically. This is described by Hudson and Parthasarathy in [11]. The operators we require act on the exponential vectors in the Boson Fock space as follows:$$A_{t}\psi \left(g\right)=\left(\int_{0}^{t}g\left(s\right)ds\right)\psi \left(g\right),A_{t}^{†}\psi \left(g\right)=\frac{d}{d\epsilon }{|}_{\epsilon =0}\psi \left(g+\epsilon \chi_{\left(0,t\right)}\right),\Lambda_{t}\psi \left(g\right)=\frac{d}{d\epsilon }{|}_{\epsilon =0}\psi \left(e^{\epsilon \chi_{\left(0,t\right)}}g\right)$$

Further we can define the stochastic differentials as:$$dA_{t}=\left(A_{t+dt}-A_{t}\right),dA_{t}^{†}=\left(A_{t+dt}^{†}-A_{t}^{†}\right),d\Lambda_{t}=\left(\Lambda_{t+dt}-\Lambda_{t}\right)$$

The significance of these operators derives from the functional form for the time development operator. In order for $U_{t}$ to be unitary, it must have the following form (see [11] Section 7):(3)$$dU_{t}=-\left(\left(iH+\frac{1}{2}L^{*}L\right)dt+L^{*}SdA_{t}-LdA_{t}^{†}+\left(1-S\right)d\Lambda_{t}\right)U_{t}$$
where H,L and *S* are bounded linear operators on H, with *H* self-adjoint, and *S* unitary. With L=0, and S=1, this reduces to the drift quantum SDE given in Equation (Equation 2).

### 2.2. Quantum Ito Formula, and Quantum Black–Scholes

The classical Black–Scholes equation is derived by first expanding the derivative valuation function V(X,t) using Ito’s lemma. Then constructing a replicating portfolio, which eliminates the risky terms, equating the 2, and assuming that the return on the original investment V(X,t) is given by the return on the chosen numeraire asset. The derivation of the quantum Black–Scholes follows similar steps. The full derivation is outlined by Accardi and Boukas in [10]. In this section we give an overview. The first step is the quantum version of the Ito formula

#### 2.2.1. Quantum Ito Formula

The quantum stochastic differentials can be combined using the following multiplication table (see [10] Lemma 1, and [11] Theorem 4.5):
$$\begin{array}{ccccc} - & dA_{t}^{†} & d\Lambda_{t} & dA_{t} & dt\\ dA_{t}^{†} & 0 & 0 & 0 & 0\\ d\Lambda_{t} & dA_{t}^{†} & d\Lambda_{t} & 0 & 0\\ dA_{t} & dt & dA_{t} & 0 & 0\\ dt & 0 & 0 & 0 & 0 \end{array}$$

We can see from the table above that:$$E[(\int_{0}^{t}dA_{s}+dA_{s}^{†}){}^{2}]=E[\int_{0}^{t}dA_{s}dA_{s}+dA_{s}^{†}dA_{s}^{†}+dA_{s}dA_{s}^{†}+dA_{s}^{†}dA_{s})]=\int_{0}^{t}ds=t=E\left[W{\left(t\right)}^{2}\right].$$

In fact, with S=1 in Equation (Equation 3), the terms in $d\Lambda_{t}$ disappear. The resulting operator is commutative, and the resulting PDE is the same as the classical Black–Scholes PDE ([10] Proposition 2). For S≠1, we have a non-commutative system, and the Black–Scholes equations have more complicated dynamics. This is discussed in the next sub-section. The key result, regarding the time development of $X^{k}$, can be obtained by application of the above multiplication rules, and is given by [10] Lemma 1:(4)$$\begin{array}{c} dj_{t}\left(X^{k}\right)=j_{t}\left(\lambda^{k-1}\alpha^{†}\right)dA_{t}^{†}+j_{t}\left(\alpha \lambda^{k-1}\right)dA_{t}+j_{t}\left(\lambda^{k}\right)d\Lambda_{t}+j_{t}\left(\alpha \lambda^{k-2}\alpha^{†}\right)dt\\ \alpha =\left[L^{*},X\right]S,\alpha^{†}=S^{*}\left[X,L\right],\lambda =S^{*}XS-X \end{array}$$

#### 2.2.2. Non-Commutative Quantum Black–Scholes

In this subsection we follow the derivation of the quantum Black–Scholes given in [10] Lemma 2. First start with the assumption that the derivative price is given by: $V_{t}=F\left(t,j_{t}\left(X\right)\right)$, and that this can be expanded as a power series: $F\left(t,x\right)=\sum_{n,k\geq 0}a_{n,k}{\left(t-t_{0}\right)}^{n}{\left(x-x_{0}\right)}^{k}$, where:$$a_{n,k}\left(t_{0},x_{0}\right)=\frac{F_{n,k}\left(t_{0},x_{0}\right)}{n!k!},\mathrm{and}:\;F_{n,k}\left(t_{0},x_{0}\right)=\frac{\partial^{n+k}F}{\partial t^{n}\partial x^{k}}{|}_{t=t_{0},x=x_{0}}.$$

Next, as in the classical case, we assume we can form a replicating strategy by investing $a_{t}$ at time *t*, in the risky asset, and the remainder of the portfolio in the numeraire asset. Thus we can equate the following:

Replicating strategy:$$dV_{t}=a_{t}dj_{t}\left(X\right)+\left(V_{t}-a_{t}j_{t}\left(X\right)\right)rdt$$

Power series expansion (ignoring terms in $\left(dt^{n}\right)$ for n≥2):$$dV_{t}=a_{1,0}\left(t,j_{t}\left(X\right)\right)+\sum_{k\geq 2}a_{0,k}\left(t,j_{t}\left(X\right)\right)j_{t}\left(X^{k}\right)$$

These 2 can be equated using Equation (Equation 4). The terms in dt yield the quantum Black–Scholes equation:(5)$$\begin{array}{c} a_{1,0}\left(t,j_{t}\left(X\right)\right)+a_{0,1}\left(t,j_{t}\left(X\right)\right)j_{t}\left(\theta \right)+\sum_{k=2}^{\infty }a_{0,k}\left(t,j_{t}\left(X\right)\right)j_{t}\left(\alpha \lambda^{k-2}\alpha^{†}\right)=a_{t}j_{t}\left(\theta \right)+V_{t}r-a_{t}j_{t}\left(X\right)r\\ j_{t}\left(\theta \right)=i\left[H,X\right]-\frac{1}{2}\left(L^{*}LX+XL*L-2L^{*}XL\right) \end{array}$$

The terms in $dA_{t}$, $dA_{t}^{†}$, and $d\Lambda_{t}$ are equated to zero by applying the following boundary conditions:$$\sum_{k=1}^{\infty }a_{0,k}\left(t,j_{t}\left(X\right)\right)j_{t}\left(\lambda^{k-1}\alpha^{†}\right)=a_{t}j_{t}\left(\alpha^{†}\right)$$
$$\sum_{k=1}^{\infty }a_{0,k}\left(t,j_{t}\left(X\right)\right)j_{t}\left(\alpha \lambda^{k-1}\right)=a_{t}j_{t}\left(\alpha \right)$$
$$\sum_{k=1}^{\infty }a_{0,k}\left(t,j_{t}\left(X\right)\right)j_{t}\left(\lambda^{k}\right)=a_{t}j_{t}\left(\lambda \right)$$

#### 2.2.3. Translations, and Classical Black–Scholes

In this subsection, based on the analysis presented in [12], we briefly summarise how by applying unitary transformations to a classical Black–Scholes system, one obtains new Quantum Black–Scholes PDEs. In the classical case, we have S=1, and therefore λ=0. Furthermore, $j_{t}\left(\alpha \alpha^{†}\right)$ is given by $\sigma^{2}x^{2}$. The boundary conditions now reduce to the single condition:$$a_{0,1}\left(t,j_{t}\left(X\right)\right)=a_{t}$$

Plugging this into Equation (Equation 5) gives: $$a_{1,0}\left(t,j_{t}\left(X\right)\right)+a_{0,1}\left(t,j_{t}\left(X\right)\right)j_{t}\left(X\right)r+a_{0,2}\sigma^{2}j_{t}\left(X^{2}\right)-V_{t}r=0$$

Using standard notation we get:(6)$$\frac{\partial V}{\partial t}+rx\frac{\partial V}{\partial x}+\frac{\sigma^{2}x^{2}}{2}\frac{\partial^{2}V}{\partial x^{2}}-rV=0$$
which is the classical Black–Scholes PDE. We now look for different unitary transformations: *S* that can be applied. The natural Hilbert space for an equity price (say FTSE price) is: $H=L^{2}\left(R\right)$. In this instance, we can look at the unitary transformations:$$T_{\varepsilon }f\left(x\right)\to f\left(x-\varepsilon \right).$$

In this case we have, for a translation invariant Lebesgue measure μ:$$\langle T_{\varepsilon }f|T_{\varepsilon }g\rangle =\int_{R}\bar{f(x-\varepsilon )}g\left(x-\varepsilon \right)d\mu \left(x\right)=\int_{R}\bar{f(x)}g\left(x\right)d\mu \left(x\right)=\langle f|g\rangle$$

Therefore, $T_{\varepsilon }$ is unitary and we get:$$\lambda f\left(x\right)=T_{-\varepsilon }XT_{\varepsilon }f\left(x\right)-Xf\left(x\right)=T_{-\varepsilon }xf\left(x-\varepsilon \right)-xf\left(x\right)=\left(x+\varepsilon \right)f\left(x\right)-xf\left(x\right)=\varepsilon f\left(x\right)$$

The resulting Quantum Black–Scholes equation is:(7)$$\frac{\partial V}{\partial t}+rx\frac{\partial V}{\partial x}+\sigma^{2}x^{2}\sum_{k\geq 2}\frac{\varepsilon^{k-2}}{k!}\frac{\partial^{k}V}{\partial x^{k}}-rV=0$$

See [12] for further discussion around the link between this PDE, and McKean stochastic processes, and Monte-Carlo simulation of solutions.

#### 2.2.4. Rotation, and Bid-Offer Interference

An alternative transformation, that can be applied in multiple dimensions, involves rotations (see [12]). For example, if *x* represents the mid-price for the FTSE, we could represent the bid-offer spread using an additional parameter: ϵ, so that (x−ϵ) represented the best bid price, and (x+ϵ) represented the best offer price. Classically, if we assume that we can trade the mid-price (for example during an end of day auction process on an electronic exchange), and the derivative contract depends only on the mid-price, then terms involving the bid-offer spread ϵ will drop out of the Black–Scholes PDE. In this case, the additional market information, provided by the level of the bid-offer spread, is irrelevant. The impact of non-zero bid-offer spread, only impacts the model once one takes into account that one cannot, in general, hedge at the mid-price (for example see [20]).

It is shown in [12] that, in the non-commutative quantum case, there is bid-offer interference that impacts the price and model dynamics, even if one can trade at the mid-price at any time. The Hilbert space is now $H=L^{2}\left(R^{2}\right)$. Further we define:$$Xf\left(x,\epsilon \right)=xf\left(x,\epsilon \right),Ef\left(x,\epsilon \right)=\epsilon f\left(x,\epsilon \right),\mathrm{and}\;S_{\phi }f\left(x,\epsilon \right)=f\left(\cos\left(\phi \right)x-\sin\left(\phi \right)\epsilon ,\cos\left(\phi \right)\epsilon +\sin\left(\phi \right)x\right).$$

In this case we end up with:$$\lambda_{x}f\left(x,\epsilon \right)=S_{\phi }^{*}XS_{\phi }f\left(x,\epsilon \right)-Xf\left(x,\epsilon \right)=\left((\cos(\phi )-1)x+\sin(\phi )\epsilon \right)f\left(x,\epsilon \right)$$
$$\lambda_{\epsilon }f\left(x,\epsilon \right)=S_{\phi }^{*}ES_{\phi }f\left(x,\epsilon \right)-Ef\left(x,\epsilon \right)=\left((\cos(\phi )-1)\epsilon -\sin(\phi )x\right)f\left(x,\epsilon \right)$$

Inserting this back into Equation (Equation 5), and making the assumption that the random variables *x* and ϵ are uncorrelated, leads to the following Quantum Black–Scholes (see [12] for details):(8)$$\begin{array}{c} \frac{\partial V(t,x,\epsilon )}{\partial t}+rx\frac{\partial V(t,x,\epsilon )}{\partial x}+r\epsilon \frac{\partial V(t,x,\epsilon )}{\partial \epsilon }-V\left(t,x,\epsilon \right)r\\ +\sigma_{x}^{2}x^{2}\sum_{k=2}^{\infty }\frac{{\left(\left(\cos\left(\phi \right)-1\right)x+\sin\left(\phi \right)\epsilon \right)}^{k-2}}{k!}\frac{\partial^{k}V\left(t,x,\epsilon \right)}{\partial x^{k}}\\ +\sigma_{\epsilon }^{2}\epsilon^{2}\sum_{l=2}^{\infty }\frac{{\left(\left(\cos\left(\phi \right)-1\right)\epsilon -\sin\left(\phi \right)x\right)}^{l-2}}{l!}\frac{\partial^{l}V\left(t,x,\epsilon \right)}{\partial \epsilon^{l}}=0 \end{array}$$

In the case ϕ=0, the terms for k,l≥3 drop out. Furthermore, if the derivative payout depends only on *x* and not ϵ, the boundary condition will be defined as a function of *x*, which will lead to a solution such that ∂V/∂ϵ=0. Thus, the case ϕ=0 leads to the classical Black–Scholes. For ϕ≠0, there will be complex dynamics even in the event of 0 correlation between the 2 variables.

For example, the choice of ϕ=π/2, leads to Equation (Equation 9), where the multipliers for the *x* partial derivatives, $\partial^{k}V/\partial x^{k}$ are dependent on the bid-offer spread ϵ. This will lead to dependence on ϵ, even where the boundary condition is a function of *x* only. While values of ϕ closer to zero are likely a better match for the real market, it does highlight the possibilities of the quantum stochastic approach.
(9)$$\begin{array}{c} \frac{\partial V(t,x,\epsilon )}{\partial t}+rx\frac{\partial V(t,x,\epsilon )}{\partial x}+r\epsilon \frac{\partial V(t,x,\epsilon )}{\partial \epsilon }-V\left(t,x,\epsilon \right)r\\ +\sigma_{x}^{2}x^{2}\sum_{k=2}^{\infty }\frac{\epsilon^{k-2}}{k!}\frac{\partial^{k}V\left(t,x,\epsilon \right)}{\partial x^{k}}+\sigma_{\epsilon }^{2}\epsilon^{2}\sum_{l=2}^{\infty }\frac{{\left(-x\right)}^{l-2}}{l!}\frac{\partial^{l}V\left(t,x,\epsilon \right)}{\partial \epsilon^{l}}=0 \end{array}$$

## 3. Nonlocal Diffusion and Quantum Stochastic Processes

### 3.1. Kolmogorov’s Forward Equation

With interest rates set to zero, the Quantum Black–Scholes equation (for example Equations (Equation 7) and (Equation 8)) becomes a standard Kolmogorov backward equation. The general form for these equations, in 2 variables, is given by:(10)$$\begin{array}{c} \frac{\partial u(t,x,\epsilon )}{\partial t}=g_{1}\left(x,\epsilon \right)\sum_{k=2}^{\infty }\frac{f_{1}{\left(x,\epsilon ,\varepsilon \right)}^{k-2}}{k!}\frac{\partial^{k}u\left(t,x,\epsilon \right)}{\partial x^{k}}\\ +g_{2}\left(x,\epsilon \right)\sum_{l=2}^{\infty }\frac{f_{2}{\left(x,\epsilon ,\varepsilon \right)}^{l-2}}{l!}\frac{\partial^{l}u\left(t,x,\epsilon \right)}{\partial \epsilon^{l}} \end{array}$$

The associated Kolmogorov forward equation, to Equation (Equation 10) is given by [12] Proposition 3.1:(11)$$\begin{array}{c} \frac{\partial p(x,\epsilon ,t)}{\partial t}=\sum_{k=2}^{\infty }\frac{{\left(-1\right)}^{k}}{k!}\frac{\partial^{k}\left(g_{1}{\left(x,\epsilon \right)}^{1/2}f_{1}{\left(x,\epsilon ,\varepsilon \right)}^{k-2}p\left(x,\epsilon ,t\right)\right)}{\partial x^{k}}\\ +\sum_{l=2}^{\infty }\frac{{\left(-1\right)}^{l}}{l!}\frac{\partial^{l}\left(g_{2}{\left(x,\epsilon \right)}^{1/2}f_{2}{\left(x,\epsilon ,\varepsilon \right)}^{l-2}p\left(x,\epsilon ,t\right)\right)}{\partial \epsilon^{l}} \end{array}$$

In Section 3 of [12], it is shown that Equation (Equation 11) can be associated with the forward equation for a nonlocal diffusion process, via a moment matching technique.

For example, in a simple translation case we set: $g_{1}\left(x,\epsilon \right)=\sigma^{2}$, $f_{1}\left(x,\epsilon ,\varepsilon \right)=\varepsilon$, and $g_{2}\left(x,\epsilon \right)=f_{2}\left(x,\epsilon ,\varepsilon \right)=0$. Equation (Equation 11) becomes:(12)$$\frac{\partial p(x,t)}{\partial t}=\sigma^{2}\sum_{k=2}^{\infty }\frac{{\left(-1\right)}^{k}\varepsilon^{k-2}}{k!}\frac{\partial^{k}\left(p\left(x,t\right)\right)}{\partial x^{k}}$$

The forward equation for a nonlocal diffusion process can be written:(13)$$\frac{\partial p(x,t)}{\partial t}=\frac{\sigma^{2}}{2}\frac{\partial^{2}}{\partial x^{2}}\left(\int_{-\infty }^{\infty }H\left(y\right)p\left(x-y,t\right)dy\right)$$

The function H(y) has the impact of “blurring” the impact of the diffusion operator. If H(y) is replaced with a dirac function δ(y), then this equation reduces to a standard Kolmogorov forward equation. Next we use the Kramers–Moyal technique of expanding about y=0: $p\left(x-y,t\right)=\sum_{k\geq 0}\frac{{\left(-1\right)}^{k}y^{k}}{k!}\frac{\partial^{k}p\left(x,t\right)}{\partial x^{k}}$. Inserting this into Equation (Equation 13) we get:(14)$$\frac{\partial p(x,t)}{\partial t}=\frac{\sigma^{2}}{2}\left(\sum_{k\geq 0}\left(\frac{{\left(-1\right)}^{k}}{k!}\frac{\partial^{k}\left(p\left(x,t\right)\right)}{\partial x^{k}}\int_{-\infty }^{\infty }y^{k}H\left(y\right)dy\right)\right)$$

Now equation coefficients between Equations (Equation 12) and (Equation 14) we get ([12], Proposition 3.2):

**Lemma** **1.**
*The Kolmogorov forward Equation (Equation 12), associated with the translation applied in Section 2.2.3, can be written as a nonlocal diffusion Equation (Equation 13). Let $H_{i}$ represent the ith moment of the function H(y). Then $H_{i}$ is given by: $H_{i}=\frac{2{\left(\varepsilon \right)}^{i}}{\left(i+1)(i+2\right)}$.*


**Proof.** The proof is obtained by equating coefficients in each $k^{th}$ partial derivative. For details refer to [12] Proposition 3.2. □

The link between quantum stochastic processes and nonlocal diffusions provides a useful tool in terms of visualising the solutions, and simulating using Monte-Carlo methods. In [12] it is shown that by writing the solution as a McKean–Vlasov process (see [13]) we can apply the particle method ([14], Chapters 10 and 11) to simulate the solution. For example, the nonlocal diffusion described by Equation (Equation 13), can be written as (see [12], Section 4):(15)$$dx=\sigma {\left(\frac{1}{p(x,t)}E^{p(y,t)}\left[H(x-y|x)\right]\right)}^{1/2}dW$$

Therefore, given the usefulness in this context, it is natural to ask whether there is a deeper link between nonlocal diffusions and the quantum stochastic processes described by Hudson and Parthasarathy. The first step is to inverse the question tackled in [12]. Given a particular nonlocal diffusion, can we write the solution as a quantum stochastic process?

### 3.2. Nonlocal Diffusion on a Riemannian Manifold

In this section, we apply a similar methodology to that outlined by Henry-Labordère in [15] Chapter 4 and 5, to a nonlocal diffusion. A general Laplacian on a 1 dimensional Riemannian manifold with metric: g(x), is given by:(16)$$\Delta_{g}=g^{-1/2}\left(\frac{\partial }{\partial x}+A_{x}\right)g^{-1/2}\left(\frac{\partial }{\partial x}+A_{x}\right)+Q\left(x\right)$$
where $A_{x}$ represents the components of an Abelian connection, and Q(x) a section of a real vector bundle over the manifold *M*. The objective now is to choose $A_{x}$, and Q(x) in order to simplify the resulting partial differential equation. If we start with the assumption that $A_{x}=Q\left(x\right)=0$, then the second derivative in Equation (Equation 13) becomes the Laplace–Beltrami operator:(17)$$\frac{\partial p(t,x)}{\partial t}=\frac{\sigma^{2}}{2\sqrt{g(x)}}\frac{\partial }{\partial x}\left(\frac{1}{\sqrt{g(x)}}\frac{\partial }{\partial x}\left(\int_{-\infty }^{\infty }H\left(y\right)p\left(t,x-y\right)dy\right)\right)$$

The next step is to apply the Kramers–Moyal expansion to Equation (Equation 17), as we did for Equation (Equation 14):(18)$$\frac{\partial p(t,x)}{\partial t}=\frac{\sigma^{2}}{2\sqrt{g(x)}}\frac{\partial }{\partial x}\left(\frac{1}{\sqrt{g(x)}}\frac{\partial }{\partial x}\left(\sum_{k\geq 0}\int_{-\infty }^{\infty }\frac{y^{k}}{k!}H\left(y\right)dy\frac{{\left(-1\right)}^{k}}{k!}\frac{\partial^{k}p\left(t,x\right)}{\partial x^{k}}\right)\right)$$

Writing: $H_{i}=\int_{-\infty }^{\infty }y^{i}H\left(y\right)dy$, and expanding out the partial derivatives in Equation (Equation 18), we get:(19)$$\frac{\partial p}{\partial t}=\frac{\sigma^{2}}{4}\frac{\partial (g^{-1})}{\partial x}H_{0}\frac{\partial p}{\partial x}+\frac{\sigma^{2}}{2}\sum_{k\geq 2}\left(\frac{{\left(-1\right)}^{\left(k-2\right)}\left(g^{-1}\right)H_{k-2}}{(k-2)!}+\frac{{\left(-1\right)}^{\left(k-1\right)}\frac{\partial (g^{-1})}{\partial x}H_{k-1}}{2(k-1)!}\right)\frac{\partial^{k}p}{\partial x^{k}}$$

To find an equivalent quantum stochastic process, we start with the relevant quantum Kolmogorov backward equation of which Equation (Equation 7) is a special type, and attempt to fit the coefficients μ(x),f(x), and ε to Equation (Equation 19):(20)$$\frac{\partial u}{\partial t}+\mu \left(x\right)\frac{\partial u}{\partial x}+f\left(x\right)\sum_{k\geq 2}\frac{\varepsilon^{k-2}}{k!}\frac{\partial^{k}u}{\partial x^{k}}=0$$

The relevant forward equation is given by (see [12] Proposition 3.1):(21)$$\frac{\partial p}{\partial t}=\mu \left(x\right)\frac{\partial p}{\partial x}+f\left(x\right)\sum_{k\geq 2}\frac{{\left(-\varepsilon \right)}^{k-2}}{k!}\frac{\partial^{k}p}{\partial x^{k}}$$

Equating coefficients for the first partial derivative we find that the quantum stochastic process we require must have a drift coefficient: $\mu \left(x\right)=\frac{\sigma^{2}}{4}\frac{\partial (g^{-1})}{\partial x}H_{0}^{g}$. Equating coefficients for higher partial derivatives we get an infinite series of 1st order partial differential equations to solve:(22)$$k\left(k-1\right)\left(g^{-1}\right)H_{k-2}-\frac{k}{2}\frac{\partial (g^{-1})}{\partial x}H_{k-1}=2f\left(x\right)\varepsilon^{k-2}/\sigma^{2}$$

By inserting the moments $H_{i}$, we end up with an infinite number of equations, where we only have the functions f(x), g(x), and the constant ε, to change. To make progress we must either simplify Equation (Equation 22), or simplify the original Laplacian. We therefore investigate the following ways of achieving this:Setting $\partial (g^{-1})/\partial x$ to zero.Setting $g{\left(x\right)}^{-1}$ to be a fixed multiple of the volatility function f(x), in Equation (Equation 21).Introducing a non-zero connection: $A_{x}$, and line bundle section: Q(x), to simplify Equation (Equation 19).

We investigate each of these in turn below.

(1) $\partial (g^{-1})/\partial x=0$:

In this case, we set $f\left(x\right)=\sigma^{2}$, g(x)=1. We are back to the standard nonlocal diffusion in Euclidean space. Equation (Equation 22) becomes:(23)$$k\left(k-1\right)H_{k-2}^{g}=2\varepsilon^{k-2}$$
and the moments of H(y) must be those given by Lemma 1.

(2) $g{\left(x\right)}^{-1}=f\left(x\right)/\sigma^{2}$:

This case now represents nonlocal diffusion on a curved manifold. However, instead of allowing a general Riemannian metric we simplify Equation (Equation 22) by restricting the functional form for g(x). If $g{\left(x\right)}^{-1}=f\left(x\right)/\sigma^{2}$, Equation (Equation 22) becomes:
(24)$$\frac{\partial (g^{-1})}{\partial x}-\left(\frac{2k\left(k-1\right)H_{k-2}-4\varepsilon^{\left(k-2\right)}}{kH_{k-1}}\right)\left(g^{-1}\right)=0$$

If the moments $H_{k}$ follow the recurrence relation, for all *k*:(25)$$\alpha =\left(\frac{2k\left(k-1\right)H_{k-2}-4\varepsilon^{\left(k-2\right)}}{kH_{k-1}}\right)$$

We get a single equation for $g{\left(x\right)}^{-1}$:
(26)$$\frac{\partial (g^{-1})}{\partial x}-\alpha g^{-1}=0$$

This equation can be solved to yield $g{\left(x\right)}^{-1}=exp\left(\alpha x\right)$, or alternatively: g(x)=exp(−αx). By assuming, without loss of generality, $H_{0}=1$, the remaining moments for H(y) follow. In fact, we have proved the following generalisation of Lemma 1, which gives a one parameter family of nonlocal diffusions on curved Riemannian manifolds, that have a natural representation as a quantum stochastic process:

**Lemma** **2.**
*Given a quantum stochastic process defined by the following Kolmogorov forward equation:*
(27)$$\frac{\partial p}{\partial t}=-\frac{\alpha \sigma^{2}exp\left(-\alpha x\right)}{4}\frac{\partial p}{\partial x}+\sigma^{2}exp\left(-\alpha x\right)\sum_{k\geq 2}\frac{{\left(-\varepsilon \right)}^{k-2}}{k!}\frac{\partial^{k}p}{\partial x^{k}}$$
*The solution can be written as a nonlocal diffusion on a Riemannian manifold with metric: g(x)=exp(−αx). The moments of the “blurring” function H(y), for α≠0 are given by the relation:*
(28)$$H_{k-1}=\frac{2k\left(k-1\right)H_{k-2}-4\varepsilon^{\left(k-2\right)}}{k\alpha }$$
*where we are free to assume $H_{0}=1$. The moments for the case α=0 reduce to the moments given for H(y) in Lemma 1.*


In the above Lemma, we have turned back to finding H(y) to fit a particular class of quantum stochastic processes. In the next section we show how, by specifying a non-zero connection: $A_{x}$, and section Q(x), we can write an associated quantum stochastic process if given the a probability density function: H(y), such that the moments exist.

(3) Solving the General Case:

Expanding out the coefficients of the general Laplacian (Equation 16), we get:(29)$$\frac{1}{g(x)}\frac{\partial^{2}}{\partial x^{2}}+\left(g{\left(x\right)}^{-1/2}\frac{\partial (g{\left(x\right)}^{-1/2})}{\partial x}+\frac{A_{x}}{g(x)}\right)\frac{\partial }{\partial x}+\left(\frac{A_{x}^{2}}{g(x)}+\frac{1}{g(x)}\frac{\partial (A_{x})}{\partial x}+Q\left(x\right)\right)$$

Therefore by choosing:(30)$$\begin{array}{c} A_{x}=-g^{1/2}\frac{\partial (g{\left(x\right)}^{-1/2})}{\partial x}\\ Q\left(f\right)=-\frac{A_{x}^{2}}{g(x)}-\frac{1}{g(x)}\frac{\partial (A_{x})}{\partial x} \end{array}$$
the Laplacian operator is simplified to:(31)$$\frac{1}{g(x)}\frac{\partial^{2}}{\partial x^{2}}$$

For a general nonlocal diffusion without drift, on a Riemannian manifold with metric g(x), the Kolmogorov forward equation can therefore be written:(32)$$\frac{\partial p(x,t)}{\partial t}=\frac{1}{g(x)}\frac{\partial^{2}}{\partial x^{2}}\left(\int_{-\infty }^{\infty }H\left(y\right)p\left(x-y,t\right)dy\right)$$

By changing coordinates to: $s=\int_{x_{0}}^{x}\frac{1}{\sqrt{g(y)}}dy$, this Kolmogorov forward equation is simplified to:(33)$$\frac{\partial p^{\prime }\left(s,t\right)}{\partial t}=\frac{1}{2}\frac{\partial^{2}}{\partial s^{2}}\left(\int_{-\infty }^{\infty }H\left(y\right)p^{\prime }\left(s-y,t\right)\frac{dy}{\sqrt{g(y)}}\right)$$
where $p^{\prime }\left(s,t\right)$ represents the probability law p(x,t) in the new coordinate system. By matching the Riemannian metric such that the moments in the new coordinate system match those from Lemma 1, we can write an associated quantum stochastic process. In other words, given a particular H(y), where the moments of H(y) exist, we match the geometry of the manifold on which the nonlocal diffusion occurs. We have the following:

**Proposition** **1.**
*Given a probability distribution H(y), such that the moments of H(y) exist, then H(y) defines a nonlocal diffusion on a Riemannian manifold, with the Kolmogorov forward Equation (Equation 33). If the Riemannian metric satisfies the set of integral equations (for i=0 to ∞):*
(34)$$\int_{-\infty }^{\infty }y^{i}H\left(y\right)\frac{dy}{\sqrt{g(y)}}=\frac{2{\left(\varepsilon \right)}^{i}}{\left(i+1)(i+2\right)}$$
*then the solution can be represented as the following quantum stochastic process:*
(35)$$j_{t}\left(X\right)=U_{t}^{*}\left(X⊗I\right)U_{t}=\sigma \left(X\right)dA_{t}^{†}+\sigma \left(X\right)dA_{t}+\varepsilon d\Lambda_{t}$$


**Proof.** Applying the multiplication rules of the Hudson–Parthasarathy stochastic calculus to Equation (Equation 35), we get the following Kolmogorov backward equation:
(36)$$\frac{\partial u(x,t)}{\partial t}+\sigma^{2}\left(x\right)\sum_{k=2}^{\infty }\frac{\varepsilon^{k-2}}{k!}\frac{\partial^{k}\left(u\left(x,t\right)\right)}{\partial x^{k}}=0$$From [12] Proposition 3.1, the associated forward equation is given by:
(37)$$\frac{\partial p(x,t)}{\partial t}=\sum_{k\geq 2}\frac{{\left(-1\right)}^{k}\varepsilon^{k-2}}{k!}\frac{\partial^{k}(\sigma^{2}\left(x\right)\left(p\left(x,t\right)\right)}{\partial x^{k}}$$We can now apply the Kramers–Moyal expansion to Equation (Equation 33) to get:
(38)$$\frac{\partial p(x,t)}{\partial t}=\frac{1}{2}\left(\sum_{k\geq 2}\int_{-\infty }^{\infty }y^{\left(k-2\right)}H\left(y\right)\frac{dy}{\sqrt{g(y)}}\frac{{\left(-1\right)}^{\left(k-2\right)}}{(k-2)!}\frac{\partial^{k}\left(\sigma^{2}\left(x\right)p\left(x,t\right)\right)}{\partial x^{k}}\right)$$The result follows by equating coefficients of each partial derivative. □

In the classical approach to finance, Henry-Labordère (in [15]) shows how the local volatility model of Dupire (see [16]) is equivalent to introducing a non-Euclidean distance metric g(x), to a Gaussian process. Equation (Equation 34) represents the same for a nonlocal diffusion, and Equation (Equation 35) for quantum diffusions.

## 4. Path Integral Approach

In this section we seek to adapt the path integral approach outlined in [4,5,6,7,8], to the Accardi–Boukas option pricing framework. This method has several benefits.

In some respects, this method is more fundamental than the conventional strategy of finding a classical Ito process that fits historical data, before solving the associated Kolmogorov backward equation. The path integral method provides a means to build the model from the underlying physical laws controlling a system via the relevant Hamiltonian function. The fact that the solution can be modelled using a Wiener process, and Gaussian kernel functions is an output of the model, rather than an input assumption. Furthermore, several different effects can be included in any model via the introduction of different potential functions. Baaquie provides an interesting application of this idea, in relation to the modelling of commodity prices, in [6].

The second key benefit relates to the fact that it suggests alternative ways to simulate quantum stochastic processes and nonlocal diffusions. A classical stochastic process can be easily simulated using random numbers generated by a Gaussian kernal. All the information required by each Monte-Carlo path is usually contained within the path itself. However, in order to apply the standard Gaussian kernel function to simulate quantum stochastic processes/nonlocal diffusions, one must write the solution as a McKean stochastic process (see Equation (Equation 15)). Each path requires information from the positions of all other paths in order to make the next step, and this therefore requires the particle method (see [14] for a description of the method, and [12] for the application to quantum stochastic processes). By deriving a non-Gaussian kernel function, one will be able to simulate the quantum process/nonlocal diffusion as a conventional stochastic process, without using the particle method.

### 4.1. First Attempt

We start with the quantum Kolmogorov backward equation:(39)$$\frac{\partial u}{\partial t}+\sigma^{2}\sum_{k\geq 2}\frac{\varepsilon^{\left(k-2\right)}}{k!}\frac{\partial^{k}u}{\partial x^{k}}=0$$

A standard approach (for example outlined in [18] Chapter 20), consists of applying a Wick rotation: t→iτ, to the Schrödinger equation:$$i\frac{\partial \psi }{\partial t}=\hat{H}\psi ,\hat{H}=\frac{{\hat{P}}^{2}}{2m},\;\mathrm{and}\;\hat{P}=-i\frac{\partial }{\partial x}$$
to get the standard heat equation:$$\frac{\partial \psi }{\partial \tau }=\frac{-1}{2m}\frac{\partial^{2}\psi }{\partial x^{2}}.$$

Applying this technique for Equation (Equation 39), we use this transformation on a Hamiltonian of the form:(40)$$\hat{H}=\frac{1}{m}\sum_{k\geq 2}\frac{\varepsilon^{k-2}{\hat{P}}^{k}}{k!},\hat{P}=-i\frac{\partial }{\partial x}$$

This leads to the following partial differential equation:(41)$$-\frac{\partial \psi }{\partial \tau }=\sum_{k\geq 2}\frac{\left(-i^{k}\right)\varepsilon^{k-2}}{k!}\frac{\partial^{k}\psi }{\partial x^{k}}$$

In addition to this equation not matching Equation (Equation 39), Hamiltonian (Equation 40) is non-Hermitian. The question then arises as to whether this situation can be rectified. In quantum mechanics, the requirement: $\hat{H}={\hat{H}}^{†}$ guarantees firstly that the spectrum of any Hamiltonian is real, and secondly that the time-evolution operator: $e^{i\hat{H}t}$ is unitary. Non-Hermitian Hamiltonians often have the problem that probability mass is not conserved. In fact, in [17] Bender shows that while the Hermiticity condition is sufficient to ensure these two requirements are met, it is not necessary. Bender shows that an alternative sufficient condition is PT symmetry. Based on the analysis from [17], we show how this applies in the current case, in the next subsection.

Most of the examples of PT symmetric quantum mechanics, discussed by Bender, involve potential terms that violate the usual Hermiticity requirement but still meet the PT symmetry requirement he derives. Hamiltonian (Equation 40) has a non-Hermitian kinetic energy term. This would cause difficulty in physical terms, for example in meeting experimentally observed relativistic invariance. However, for probability modelling it is a viable model.

### 4.2. PT Symmetric Quantum Mechanics

With ε=0 in (Equation 40), we have $H=H^{†}$, where † represents the Dirac Hermitian conjugate, and we can use conventional quantum mechanics based on the inner product defined by:$$\langle \psi |\phi \rangle =\int \psi {\left(x\right)}^{*}\phi \left(x\right)dx.$$

Let P represent a space reflection, and T a time reflection, and let $H^{\mathrm{PT}}$ represent the Hamiltonian under a combined space and time reflection. Since for ε≠0 we no longer have: $H=H^{†}$, the question arises as to whether we can use the condition $H=H^{\mathrm{PT}}$ instead. The basic requirements to build a viable quantum mechanics on this condition are:(i)Real and Symmetric Hamiltonian ($H=H^{\mathrm{PT}}$).(ii)An inner product that defines a positive norm: ${\left||\psi |\right|}^{2}=\langle \psi |\psi \rangle$.(iii)A unitary time development operator.

The analysis outlined in [17], shows how to do this. For (i), it is enough to show that the Hamiltonian operator $\hat{H}$, and PT share a common set of eigenfunctions (see [17] Section II). For a linear operator: $\hat{A}$, this condition is met if $\hat{A}$ commutes with $\hat{H}$: $[\hat{A},\hat{H}]=0$. However, in this case PT is nonlinear. Therefore, we need to find a linear operator: C that commutes with both $\hat{H}$, and PT. We can then write:(42)$${\langle \psi |\phi \rangle }^{\mathrm{CPT}}=\int \psi^{\mathrm{CPT}}\left(x\right)\phi \left(x\right)dx=\int C\psi^{*}\left(-x\right)\phi \left(x\right)dx$$

To find C, following [17], we write: $C=e^{Q(\hat{x},\hat{p})}P$, and use the fact that any $Q(\hat{x},\hat{p})$ that meets the following conditions, will meet our requirements:[C,PT]=0[C,H]=0${\langle \psi |\psi \rangle }^{\mathrm{CPT}}>0$

In our case, any function: $Q\left(\hat{x},\hat{p}\right)=f\left(\hat{p}\right)$, will commute with $\hat{H}$ given by (Equation 40), and PT. Furthermore, in this case it is sufficient for $C^{2}=1$ for the third requirement. Therefore, we can choose C=P, and (Equation 42) becomes:(43)$${\langle \psi |\phi \rangle }^{\mathrm{CPT}}=\int \psi^{\mathrm{CPT}}\left(x\right)\phi \left(x\right)dx=\int \psi^{*}\left(x\right)\phi \left(x\right)dx$$

It should be noted at this point that where there is a non-zero potential function in the Hamiltonian:(44)$$\hat{H}=\frac{1}{m}\sum_{k\geq 2}\frac{\varepsilon^{k-2}{\hat{P}}^{k}}{k!}+V\left(\hat{X}\right)$$

Then, the PT symmetry requirements will place restrictions on the function $V(\hat{X})$, and lead to different choices for the operator C. For example, if $V(\hat{X})$ is a real function of $\hat{X}$, then it must be an even function, in order to maintain PT symmetry. For further details, we refer to [17].

### 4.3. Second Attempt

Now that we are confident that the non-Hermitian Hamiltonian given by Equation (Equation 40) does lead to a viable quantum mechanics based on PT symmetry, we return to the question around how to construct a path integral. The fact that the Hamiltonian is symmetric under combined space and time reflections, rather than time alone or space alone, suggests that our Wick rotation must be extended to the space dimension, as well as the time dimension. Therefore, we try: (s,τ)=(ix,it). Under this new rotation, the Schrödinger equation:(45)$$i\frac{\partial \psi }{\partial t}=\sum_{k\geq 2}\frac{\left(-i^{k}\right)\varepsilon^{\left(k-2\right)}}{k!}\frac{\partial^{k}\psi }{\partial x^{k}}$$
is transformed to the original quantum Kolmogorov backward equation as required:(46)$$\frac{\partial \psi }{\partial \tau }+\sum_{k\geq 2}\frac{\varepsilon^{\left(k-2\right)}}{k!}\frac{\partial^{k}\psi }{\partial s^{k}}=0$$

We can now proceed with the construction of the path integral, and calculation of the required kernel function.

It is worth highlighting at this point, that in the above analysis, we have made the assumption ℏ=1, in translating the usual observables from physics. In quantum physics, this parameter defines the degree to which the non-commutativity of quantum mechanics impacts real observations. Algebraically speaking, when using Hamiltonian (Equation 40) for probability modelling, this parameter will be absorbed into the volatility σ and the parameter ε. The relative impact of the quantum non-commutativity in our case is measured by the ratio: $\frac{\varepsilon }{\sigma^{2}\delta t}$. This is discussed further below.

### 4.4. Fourier Transform Method

Following the usual steps (for example outlined in [18] Chapter 20), we start by assuming a solution to the Schrödinger equation: ψ(x,t)=exp(i(px−ω(p)t)), and insert this into:(47)$$i\frac{\partial \psi }{\partial t}=\sum_{k\geq 2}\frac{{\left(-i\right)}^{k}\varepsilon^{k-2}}{k!m}\frac{\partial^{k}\psi }{\partial x^{k}}$$

We obtain:(48)$$\omega \left(p\right)=i\sum_{k\geq 2}\frac{\varepsilon^{k-2}p^{k}}{k!m}$$

Inserting the normalising constant, we can write the wave function at *t* as:(49)$$\psi \left(x,t\right)=\frac{1}{\sqrt{2\pi }}\int_{R}\tilde{\psi_{0}}\left(p\right)exp\left(i(px-\sum_{k\geq 2}\frac{\varepsilon^{k-2}t}{k!m}p^{k})\right)dp$$

Taking the inverse Fourier transform, enables us to define an non-Gaussian integral for the path integral approach:(50)$$K_{t}^{\varepsilon }\left(x\right)=F^{-1}\left(exp\left(\frac{-it}{m}\sum_{k\geq 2}\frac{\varepsilon^{k-2}p^{k}}{k!}\right)\right)$$
where for ε=0, $K_{t}^{0}\left(x\right)=\sqrt{\left(\frac{m}{2\pi it}\right)}exp\left(\frac{imx^{2}}{2t}\right)$. Since we have been working in momentum space, the kernel function is defined on arbitrary input *x*, by taking the inverse Fourier transform. However, we are still tracking the input “parameter” time: *t*. Replacing τ=it, in the ε=0 case, leads to the usual Wiener measure.

### 4.5. Legendre Transform Method

In this section we show how to derive the required integral kernel using the standard Legendre transformation method (for example, see [19] Chapter 8). In addition to providing further insight, this has the benefit that it results in a power series solution without requiring the inverse Fourier transform. We assume the existence of “generalised” states $|x_{0}\rangle$ for a particle with position $x_{0}$, and wave-function given by $\psi \left(x\right)=\delta \left(x_{0}-x\right)$. Furthermore, we assume the existence of a generalised momentum state: $|p_{0}\rangle$, with wave-function given by $\psi \left(x\right)=e^{ip_{0}x}$. As noted in [19], it is possible to apply the theory of distributions in making this argument rigorous, although in this section we proceed on a formal basis.

For a system with Hamiltonian $\hat{H}$ and in generalised state $|x_{1}\rangle$, the probability of finishing in generalised state $|x_{2}\rangle$, after time δt is given by: $\langle x_{2}|exp\left(-i\delta t\hat{H}\right)|x_{1}\rangle$. We now proceed by inserting Hamiltonian (Equation 40) and then expanding over momentum states as follows:(51)$$\langle x_{2}|exp\left(-i\delta t\hat{H}\right)|x_{1}\rangle =\int_{R}dp\left(exp\left(-i\delta t\sum_{k\geq 2}\frac{\varepsilon^{\left(k-2\right)}p^{k}}{k!m}\right)\langle x_{2}|p\rangle \langle p|x_{1}\rangle \right)$$

We have: $\langle x_{2}|p\rangle =\frac{1}{\sqrt{2\pi }}\int_{R}\delta \left(x_{2}-x\right)e^{ipx}dx$, and $\langle p|x_{1}\rangle =\frac{1}{\sqrt{2\pi }}\int_{R}e^{-ipx}\delta \left(x_{1}-x\right)dx$, so $\langle x_{2}|p\rangle \langle p|x_{1}\rangle =\frac{1}{\sqrt{2\pi }}e^{ip(x_{2}-x_{1})}$.

Feeding this back into (Equation 51), we get:(52)$$\langle x_{2}|exp\left(-i\delta t\hat{H}\right)|x_{1}\rangle =\frac{1}{2\pi }\int_{R}dp\left(exp\left(i\left(p\left(x_{2}-x_{1}\right)-\delta t\sum_{k\geq 2}\frac{\varepsilon^{\left(k-2\right)}p^{k}}{k!m}\right)\right)\right)$$

Assuming, we have a stochastic process that makes *N* steps as outlined above, each with time-step δt, we get:(53)$$\langle x_{N},t_{N}|x_{0},t_{0}\rangle =\int_{R^{N}}\prod_{j=1}^{N-1}dx_{j}\left(\langle x_{N}|e^{-i\hat{H}\delta t}|x_{N-1}\rangle \langle x_{N-1}|e^{-i\hat{H}\delta t}|x_{N-2}\rangle \ldots \langle x_{1}|e^{-i\hat{H}\delta t}|x_{0}\rangle \right)$$

We now take the limit N→∞, and δt→0. To do this we require: $Dx=\lim_{N\to \infty }\prod_{j=1}^{N}dx_{j}$, and $Dp=\lim_{N\to \infty }\prod_{j=1}^{N}dp_{j}$, where the integral is over all paths from the initial, to the final states. It is possible (for example see [21]) to give this integral a rigorous definition, and prove convergence. However, as noted above, we proceed on a purely formal basis. Inserting (Equation 52) into (Equation 53), and taking the limit we get:(54)$$\langle x_{N},t_{N}|x_{0},t_{0}\rangle =\frac{1}{2\pi }\int DxDp\left(exp\left(i\int_{t_{0}}^{t_{N}}dt\left(p\left(t\right)\dot{x}\left(t\right)-\hat{H}\left(x,p\right)\right)\right)\right)$$
where we have introduced the notation: $\dot{x}=\frac{dx}{dt}$.

To carry out the momentum integral, we use the method of stationary phase (see for example [22] Section 6.5). This enables us to develop the central result for this section—a Feynman–Kac formula for the Quantum Kolmogorov backward equation given by Equation (Equation 39).

**Proposition** **2.**
*If u(x,t) is a solution of the quantum Kolmogorov backward equation, given by (Equation 39), then we have:*
(55)$$E[u\left(x,T\right)|u\left(x_{0},0\right)]=\int_{-\infty }^{\infty }u\left(x,0\right)K_{T}^{\varepsilon }\left(x\right)dx$$
*where $K_{T}^{\varepsilon }\left(x\right)$ is given by: *
(56)$$\begin{array}{c} K_{T}^{\varepsilon }\left(x\right)=\int_{-\infty }^{\infty }exp\left(-A(\dot{x})\right)Dx\\ A\left(\dot{x}\right)=\int_{0}^{T}\sum_{k\geq 0}\frac{{\left(-\varepsilon \right)}^{k}{\left(\dot{x}\right)}^{\left(k+2\right)}}{\sigma^{2(k+1)}\left(k+1\right)\left(k+2\right)}dt \end{array}$$
*Dx is given by:*
(57)$$Dx=C_{0}\lim_{N\to \infty }\prod_{j=1}^{N}dx_{j}I\left(\sigma^{2}\delta t,\varepsilon ,\dot{x}\right)$$
*$I(\sigma^{2}\delta t,\varepsilon ,\dot{x}))$ represents the integral:*
(58)$$I\left(\sigma^{2}\delta t,\varepsilon ,\dot{x}\right)=\int_{-\infty }^{\infty }dp\left(\exp\left(-\sigma^{2}\delta te^{\varepsilon p_{0}\left(\dot{x}\right)}\sum_{k\geq 2}\frac{\varepsilon^{k-2}{\left(p-p_{0}\left(\dot{x}\right)\right)}^{k}}{k!}\right)\right)$$
*where $C_{0}$ represents a normalisation constant. For $\left(\frac{\varepsilon }{\sigma^{2}\delta t}\right)$ small, we have:*
(59)$$Dx∼C_{0}\lim_{N\to \infty }\prod_{j=1}^{N}\frac{dx_{j}}{\sqrt{\sigma^{2}\delta t\left(1+\frac{\varepsilon \dot{x}}{\sigma^{2}}\right)}}$$


**Proof.** We start with the integral (Equation 54), in *N* discrete steps:
(60)$$\frac{1}{2\pi }\int_{-\infty }^{\infty }\prod_{i=1}^{N}dx_{i}\prod_{i=1}^{N}dp_{j}\left(exp\left(i\sum \delta t\left(h(p)\right)\right)\right)$$
where $h\left(p\right)=p\dot{x}-\sigma^{2}\sum_{k\geq 2}\frac{\varepsilon^{\left(k-2\right)}p^{k}}{k!}$. We now require the Legendre transformation, by first finding the stationary point $p_{0}$ such that $h^{\prime }\left(p_{0}\right)=0$.
(61)$$h^{\prime }\left(p\right)=\frac{\delta x_{i}}{\delta t}-\sigma^{2}\sum_{k\geq 2}\frac{\varepsilon^{\left(k-2\right)}p^{\left(k-1\right)}}{(k-1)!}$$Therefore, we get:
(62)$$p_{0}=\frac{ln(\frac{\varepsilon \dot{x}}{\sigma^{2}}+1)}{\varepsilon }$$The Legendre transformation for H(p) can therefore be written:
(63)$$\begin{array}{c} L\left(\dot{x}\right)=p_{0}\left(\dot{x}\right)\dot{x}-\sigma^{2}\sum_{k\geq 2}\frac{\varepsilon^{\left(k-2\right)}p_{0}{\left(\dot{x}\right)}^{k}}{k!}\\ p_{0}\left(\dot{x}\right)=\frac{ln(\frac{\varepsilon \dot{x}}{\sigma^{2}}+1)}{\varepsilon } \end{array}$$We now expand h(p) about the stationary point, in (Equation 60):
(64)$$\frac{1}{2\pi }\int_{-\infty }^{\infty }\prod_{i=1}^{N}dx_{i}\prod_{i=1}^{N}dp_{j}\left(exp\left(i\delta t\left(h\left(p_{0}\right)+\sum_{k}\frac{{\left(p-p_{0}\right)}^{k}}{k!}\frac{\partial^{k}h}{\partial p^{k}}{|}_{p=p_{0}}\right)\right)\right)$$From above, for k≥2, $\frac{\partial^{k}h}{\partial p^{k}}=\varepsilon^{k-2}e^{\varepsilon p_{0}}$, and $e^{\varepsilon p_{0}}=1+\frac{\varepsilon \dot{x}}{\sigma^{2}}$. Inserting this into integral (Equation 64), and making the usual Wick rotation, gives:
(65)$$\frac{1}{2\pi }\int_{-\infty }^{\infty }\prod_{i=1}^{N}dx_{i}I\left(\sigma^{2}\delta t,\varepsilon ,\dot{x}\right)\left(exp\left(-\delta t\left(h(p_{0})\right)\right)\right)$$
where:
(66)$$I\left(\sigma^{2}\delta t,\varepsilon ,\dot{x}\right)=\int_{-\infty }^{\infty }dp\left(\exp\left(-\sigma^{2}\delta te^{\varepsilon p_{0}\left(\dot{x}\right)}\sum_{k\geq 2}\frac{\varepsilon^{k-2}{\left(p-p_{0}\left(\dot{x}\right)\right)}^{k}}{k!}\right)\right)$$
as required. For $\varepsilon <\sigma^{2}\delta t$, we can approximate $I(\sigma^{2}\delta t,\varepsilon ,\dot{x})$ using the standard Gaussian integral:
(67)$$I\left(\sigma^{2}\delta t,\varepsilon ,\dot{x}\right)∼\sqrt{\frac{\pi }{\sigma^{2}\delta te^{\varepsilon p_{0}}}}$$The Lagrangian is given by:
(68)$$L\left(\dot{x}\right)=p_{0}\dot{x}-\sigma^{2}\sum_{k\geq 2}\frac{\varepsilon^{\left(k-2\right)}p_{0}^{k}}{k!}$$We can write this as:
(69)$$L\left(\dot{x}\right)=\frac{1}{\varepsilon }ln\left(\frac{\varepsilon \dot{x}}{\sigma^{2}}+1\right)\dot{x}-\frac{\sigma^{2}}{\varepsilon^{2}}\left(e^{\varepsilon p_{0}}-\varepsilon p_{0}-1\right)$$Expanding in escalating powers of ε we find:
(70)$$L\left(\dot{x}\right)=\sum_{k\geq 0}\frac{{\left(-\varepsilon \right)}^{k}{\left(\dot{x}\right)}^{\left(k+2\right)}}{\sigma^{2(k+1)}\left(k+1\right)\left(k+2\right)}$$Pulling everything together, and taking the infinitesimal limit (established in [21]) we have:
(71)$$\begin{array}{c} \langle x_{N},t_{N}|x_{0},t_{0}\rangle =\int Dx\left(exp\left(i\int_{t_{0}}^{t_{N}}dt\left(L(\dot{x})\right)\right)\right)\\ L\left(\dot{x}\right)=\sum_{k\geq 0}\frac{{\left(-\varepsilon \right)}^{k}{\left(\dot{x}\right)}^{\left(k+2\right)}}{\sigma^{2(k+1)}\left(k+1\right)\left(k+2\right)}\\ Dx∼\lim_{N\to \infty }\prod_{j=1}^{N}dx_{j}{\left(\frac{C_{0}}{2\pi \sigma^{2}\delta t\left(1+\varepsilon \dot{x}\right)}\right)}^{1/2} \end{array}$$After applying the usual Wick rotation, Equation (Equation 71) gives us the probability density for *x* that can be used to derive the solution: u(x,t) required (where $C_{0}$ is a normalisation constant). □

The form of the Lagrangian in Proposition 2, illustrates the fact that the strength of the quantum interaction is driven by the ratio: $\frac{\varepsilon }{\sigma^{2}\delta t}$. In this case the Lagrangian includes the usual quadratic term in $\frac{{\left(\delta x\right)}^{2}}{\sigma^{2}\delta t}$. The higher order terms have the multiplier: $\frac{{\left(-\varepsilon \right)}^{k}}{{\left(\sigma^{2}\delta t\right)}^{k+1}}$, increasing by a factor of $\frac{\varepsilon }{\sigma^{2}\delta t}$ with each increase in *k*. Therefore, where this ratio is <<1, the impact of the quantum interaction is expected to be small, and the process will be well approximated by a classical process.

## 5. Numerical Results and Future Work

### 5.1. Numerical Results

Numerical simulation of quantum stochastic processes is challenging, due to the singular nature of the partial differential equations governing their behaviour. In [12], the author outlines how the Particle method may be used to simulate their solutions. In this paper, we have shown how to develop a kernel function that can be used to numerically integrate solutions, although the accurate calculation of the kernel function is itself challenging. In this section we briefly illustrate some of the key behaviours observed.

Figure 1 below shows the kernel function given by Proposition 2, for a 1 day simulation of a process with σ=0.2.

The green curve represents a standard Gaussian process (ε=0). We show the results for ε=±0.005, and ±0.01. Introducing a non-zero translation has had the effect of introducing skew to the distribution, as well as increasing the kurtosis (so called “fat tails”). The yellow/orange curves show the results with translation ε=±0.01, and the effect is the strongest. These results are consistent with the results, obtained using the particle method, from [12]. There are many means to introduce such effects into the simulation of traded market prices. For example via classical local volatility (see for example [16]), or stochastic volatility (see for example [23]) models. The key difference in this case is that effects such as downside skew, and “fat tails" observed for real in the market, are an output of simple physical assumptions, rather than introduced exogenously. The behaviours of the market are reflected in the Hamiltonian function, and the resulting integral kernel, rather than applied via the choice of the volatility parameter.

Figure 2 shows the same kernels after a 1 year time period. As noted above, the ratio $\frac{\varepsilon }{\sigma^{2}\delta t}$ is a measure of the degree to which the quantum effects will be observed. As the time parameter *t* tends to infinity, the kernel function will asymptotically tend to a Gaussian kernel. In fact this reproduces another problematic observation of the behaviour of real financial markets, namely the flattening of the forward skew.

The local volatility model shows how to fit a full probability distribution to the current Black–Scholes option smile (prices of vanilla put and call options at different strikes and maturities). Asymptotic analysis (for example see [15] Chapter 5) can be used to show that these probability distributions derived from real market prices of vanilla options show a persistent flattening of the short maturity skew, as one moves forward in time. In fact, there are theoretical no-arbitrage arguments why the skew must asymptotically approach zero for long maturities outlined in [24] Section 5. Despite this, the skew flattening is not observed in reality—short maturity skew for options does not flatten. This issue causes problems in the pricing of path dependent options, and is sometimes solved through introducing a stochastic component to the local volatility. Not only does the quantum stochastic process have a natural skew, resulting from the Hamiltonian governing the system, but the large short maturity skew naturally persists every time is reset to zero, and flattens for longer maturities. The rate of flattening being described by the parameter ε.

### 5.2. Future Developments

#### 5.2.1. Nonlocal Diffusions

The results presented in Section 3, and those in [12], show that there is a link between Hudson–Parthasarathy quantum stochastic processes, and nonlocal diffusions specified by a convolution function: H(y) when all the moments of H(y) exist. However, the link established is purely “coincidental” in the sense that no physical link between the 2 different types of diffusion is given. The nonlocal diffusion enters the picture purely through its Kramers–Moyal expansion. One avenue of future theoretical work would be to develop the integral version of the quantum Kolmogorov equations from physical principals, and to tie this into the theory of quantum stochastic processes. One natural question to ask, is to whether there is a quantum representation of all nonlocal diffusion processes (including those where the moments of H(y) are not defined), and if not whether there is a physical (or financial) explanation for this.

#### 5.2.2. Multi-Dimensional Quantum Stochastic Processes

In one dimension, the methodology outlined provides an interesting, and natural, way of including market effects such as the observed “skew” in implied volatility, and the phenomenon of “fat tails” into the kernel function used to price derivatives. Indeed, the parameter ε has a natural understanding as a market “fear factor”. However, it is in multiple dimensions where the true benefits of the quantum approach become apparent. A traded price is not really a one dimensional variable, but consists of multiple bids and offers sitting on an exchange. Including these added pieces of information does not necessarily impact the price of a derivative using a classical model based on Ito calculus. However, in the quantum case, not only does the added dimensionality increase the variety of different quantum models available, but the different dimensions interact and there are quantum effects that cannot be replicated using a classical model. Extending the path integral methodology to these multi-dimensional models is an important next step.

#### 5.2.3. Iterated Quantum Stochastic Processes

Starting with a zero drift quantum stochastic process, where the time development operator is given by:(72)$$dU_{t}=-\left(\left(\frac{LL^{*}}{2}\right)dt+L^{*}SdA_{t}-LdA_{t}^{†}+\left(1-S\right)d\Lambda_{t}\right)U_{t}$$

When L=σ, and *S* represents a Lebesgue invariant translation: $T_{\varepsilon }$, we have shown that the system can be modelled using a Hamiltonian function given by:(73)$$\hat{H}=\sigma^{2}\sum_{k\geq 2}\frac{\varepsilon^{k-2}{\hat{P}}^{k}}{k!},\hat{P}=-i\frac{\partial }{\partial x}$$

A quantum state, evolving deterministically under this Hamiltonian, is expected to show the random behaviour discussed in Section 5.1. The next natural step is to include this Hamiltonian as the drift in Equation (Equation 72) to generate a new quantum stochastic process. In theory, this cycle could be repeated indefinitely:Choose a quantum stochastic process.Identify the relevant Hamiltonian function, by which the path integral method can be applied.Feed back the Hamiltonian into a new quantum stochastic process as the quantum drift.Go back to step 1.

This is an interesting area of possible future research.

## 6. Conclusions

In this article we have shown that, using Riemannian geometry, the representation of a quantum stochastic process as a nonlocal diffusion, can be inversed for the cases where the convolution function that defines the nonlocality (H(y)) has defined moments: $E^{H(y)}\left[y^{n}\right]$ for all *n*. We go on to show how one can use PT symmetric quantum mechanics to develop a path integral formulation of the Accardi–Boukas quantum Black–Scholes framework.

## Figures and Tables

**Figure 1 entropy-21-00105-f001:**
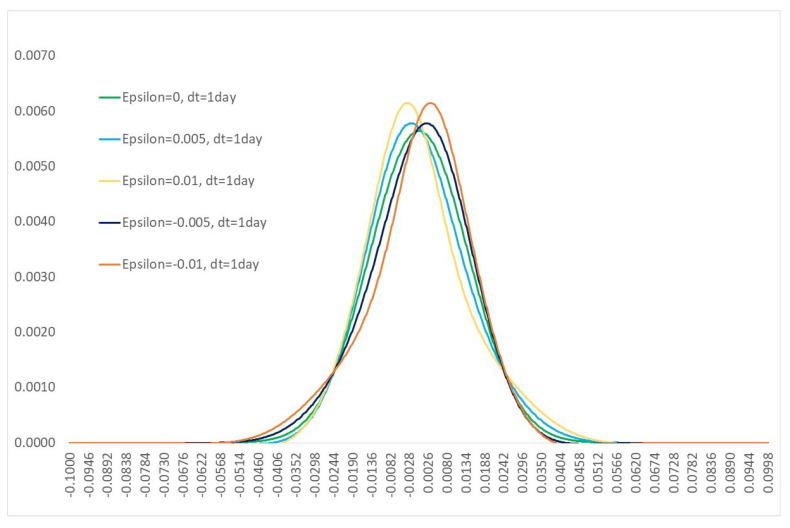
Kernel function with σ=0.2 after a 1 day time interval.

**Figure 2 entropy-21-00105-f002:**
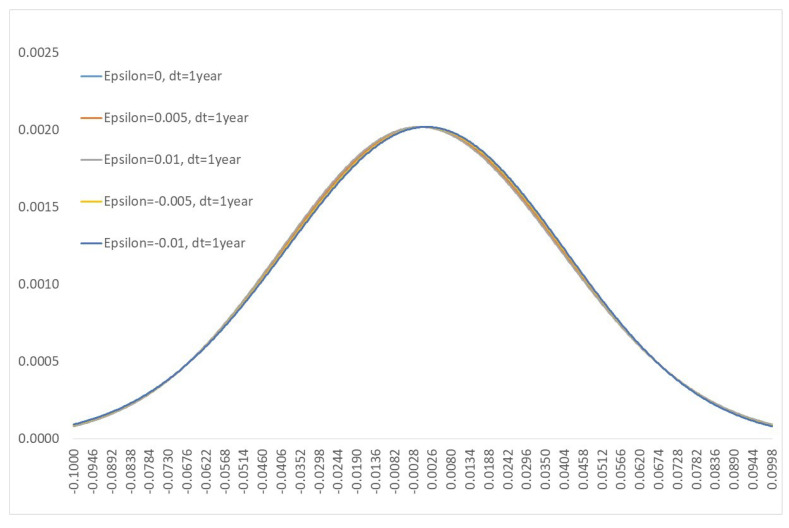
Kernel function with σ=0.2 after a 1 year time interval.

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
