# Peer review of "PT Symmetry, Non-Gaussian Path Integrals, and the Quantum Black–Scholes Equation"

_entropy, 2019, doi:10.3390/e21020105_

Round 1
Reviewer 1 Report
I read the paper by Dr. Hicks on the quantum Black-Scholes equation with great interest.
The key issue this paper answers is whether there is a link between non-local diffusions and quantum stochastic processes. For a given non-local diffusion the author finds that in certain cases, an equivalent stochastic process can be found. This is a powerful result. Starting from the Accardi-Boukas approach where a quantum stochastic process is the starting point, this paper inverses that starting point (starting from equation (14)). Already the quantum stochastic approach to derive a Black-Scholes model was a milestone. This paper now, in fact generalizes option pricing even further. This paper continues to look at the Accardi and Boukas approach also from a least action point of view. Other work already exists in the area of least action in option pricing (not connected to Accardi and Boukas). But the least action approach does away with the typical finance assumption that a Wiener process needs assuming from the start.
A poignant result of this paper is the backward Kolmogorov pde - in the quantum style, which when written as a Schrodinger PDE leads to non-hermiticity of the Hamiltonian. Of course, from a quantum mechanics point of view this means trouble. But maybe this is less so the case when one wants to use quantum formalisms in social science? Maybe the author can make some mention of this? For instance, (3) may then not be necessary? Also -may there be also some financial argument made why specifically non hermiticity obtains? Would it be linked to the non-arbitrage condition? I have a somewhat (unfair) other question: would the author know what possible equivalent finance interpretation we could give to h=1?
A very interesting observation the author makes is the one about bid-ask interference. It would be great if we could – financially speaking – attempt to fine tune this interference. The parameter epsilon is so key throughout the paper. May it be possible to highlight this?
This referee is not too clear what it means to set f(x) (in (20)) to a multiple of g(x)**-1 (line 216). Can the financial intuition be a little explained?
Can the author also give a little – financial intuition on (33)?
I think the author should explicitly highlight that market behavior is now triggered by a Hamiltonian (in reference to the result in section 5). I think this needs to be mentioned in the introduction of the paper. It is very important.
Author Response
Many thanks for taking the time to review the article. I found the feedback most helpful.
I have listed changes to the article, and responses to each point in the attached pdf.
Many Thanks
Will

Reviewer 2 Report
Quantum Stochastic Calculus is 35 years old but it is with the recent work of Will Hicks that I am seeing some solid applications. I recommend continuing on this path of research, related to the Accardi-Boukas Quantum Black-Scholes equation, with possible expansion of the numerical simulations part.
The style of writing and overall exposition is very good. As far as I can tell, the results are mathematically correct and of definite interest.
Author Response
Many thanks for taking the time to review the article. Following on from the review process, I have made some changes to the article, which I have listed, and discussed in the attached pdf.
Thanks again
Will
